# A Brief Review of the Status of Low-Pressure Membrane Technology Implementation for Petroleum Industry Effluent Treatment

**DOI:** 10.3390/membranes12040391

**Published:** 2022-03-31

**Authors:** Kasro Kakil Hassan Dizayee, Simon J. Judd

**Affiliations:** 1College of Engineering, Salahaddin University-Erbil, Erbil 44002, Iraq; kasro.dizayee@su.edu.krd; 2Cranfield Water Science Institute, Cranfield University, Bedford MK43 0AL, UK

**Keywords:** produced water, refinery effluent, petrochemical effluent, ultrafiltration, microfiltration, membrane bioreactors, ceramic membranes

## Abstract

Low-pressure membrane technology (ultrafiltration and microfiltration) has been applied to two key effluents generated by the petroleum industry: produced water (PW) from oil exploration, a significant proportion being generated offshore, and onshore refinery/petrochemical effluent. PW is treated physicochemically to remove the oil prior to discharge, whereas the onshore effluents are often treated biologically to remove both the suspended and dissolved organic fractions. This review examines the efficacy and extent of implementation of membrane technology for these two distinct applications, focusing on data and information pertaining to the treatment of real effluents at large/full scale. Reported data trends from PW membrane filtration reveal that, notwithstanding extensive testing of ceramic membrane material for this duty, the mean fluxes sustained are highly variable and generally insufficiently high for offshore treatment on oil platforms where space is limited. This appears to be associated with the use of polymer for chemically-enhanced enhanced oil recovery, which causes significant membrane fouling impairing membrane permeability. Against this, the application of MBRs to onshore oil effluent treatment is well established, with a relatively narrow range of flux values reported (9–17 L·m^−2^·h^−1^) and >80% COD removal. It is concluded that the prospects of MBRs for petroleum industry effluent treatment are more favorable than implementation of membrane filtration for offshore PW treatment.

## 1. Introduction

The challenge imposed by effluents produced by the petroleum industry has been extensively reviewed [1,2,3,4,5,6]. Effluents range from “produced water” (PW) [2,3,4,5,6] generated from oil exploration—by far the largest-volume effluent across the sector—to those relating to oil refining (i.e., the separation of crude oil into useful fractions) and the synthesis of petrochemicals [1,2]. Produced water (PW) is water that is extracted from an oil well with the crude oil during crude oil production. It results from displacing oil in the reservoir with environmental waters—most often seawater—and contains many chemical species, which are onerous to the environment and/or challenging to oil abstraction [7,8]. They include cations which form scales, such as calcium and magnesium, and toxic species such as hydrogen sulfide and heavy metals. These species, which can vary significantly in concentration both regionally and temporally, can be categorized according to their origin and/or chemistry (Figure 1). The extent to which they must be removed is determined by:(a).the legislated limits for discharge to the environment (i.e., to sea),(b).the contaminant level limits demanded by reuse of the treated effluent for reinjection into the reservoir (known as produced water reinjection, or PWRI), and/or(c).the overall wastewater management strategy and treatment technologies selected, determined largely by whether the installation is based onshore or offshore (Figure 2).

Key onshore petroleum effluents comprise those discharged from refining and petrochemical manufacturing. Refining refers to the fractionation of crude oil primarily into transportation fuels, along with heating oils and other more minor oil elements. Petrochemicals are the intermediates used to produce industrial and consumer products (e.g., plastics, rubbers, resins, synthetic fibers, adhesives, dyes, detergents and pesticides).

For such on-shore installations, where footprint is a less critical factor, relatively low-energy/high-footprint technologies can be applied. These are often based on biological treatment processes, frequently applied to industrial effluents including refinery and petrochemical effluents. For offshore applications, where space is limited [2,3,4,5,6], biological treatment is not considered feasible. The most widely applied treatment technologies on oil platforms are hydrocyclones followed by induced gas flotation (IGF) [5,6], with media filtration or other polishing technologies if it is considered essential or viable (Figure 2).

Within the petroleum sector, low-pressure (or “porous”) membrane technologies have been extensively studied based on abiotic [9,10,11,12,13,14,15,16,17,18,19,20,21,22,23], i.e., conventional perm-selective physical separation, and biological [24,25,26,27,28,29] treatment focused on membrane bioreactor (MBR) technology (Figure 3), which provides an enhancement over conventional biological treatment. Both polymeric and, increasingly for PW applications, ceramic [9,11,12,14,15,16,17,18,19,20,21,22,23,30] ultrafiltration (UF) and microfiltration (MF) materials have been tested. Although there has been a preponderance of bench-scale investigations using analogue wastewaters [9,10,11,19,21,23], it is the data from studies based on real effluents—including full-scale installations [24,25,26]—which offer the most pertinent insight into process technical performance.

Given that both on-shore and off-shore installations are faced with similar challenges in terms of the nature of the effluent, ostensibly from the biorefractory organic matter and oil content, it is of interest to compare the overall performance of the abiotic and biological processes with reference to:the influent water quality,system hydraulics, and specifically sustainability of flux and permeability (the flux per unit transmembrane pressure, TMP),organic carbon removal, as represented by the oil and/or chemical oxygen demand (COD), andrequirement for supplementary system components, and specifically pretreatment and post-treatment.

The above facets are reviewed, based on all accessible information (peer-reviewed literature, conference presentations, company reports and other grey literature). The collated and synthesized data are subsequently used to inform an assessment of the current and future likely implementation of the two membrane technologies.

## 2. Water Quality

Different treated water quality objectives apply to PW for sea discharge and land-based petroleum effluents. For the former, the discharged water quality is based on the oil concentration, and physicochemical removal of the suspended oil is normally sufficient to meet the requirement. The required water quality for effluents discharged inland, on the other hand, is normally based on the total organic concentration measured as the COD or BOD.

### 2.1. Produced Water (PW)

The ranges of concentration of the key PW bulk water quality parameters (Table 1) vary widely according to the reservoir formation and other geological characteristics [2,3,4,5,6]. PW is highly saline and is usually supersaturated in key scalant cations such as calcium, magnesium and barium. It often contains elevated levels of toxic metals such as lead. Scalant species present a challenge from precipitation within the reservoir formation pores, reducing its permeability and impeding the extraction of oil.

The organic fraction derives largely from the oil itself, which is partitioned between the dissolved/emulsified and suspended (or “free”) fractions. Free oil presents a greater challenge than dissolved or emulsified oil due to its high membrane fouling propensity. The organic chemicals making up the oil fraction are classified according to the general molecular chemistry (aliphatic or aromatic, saturated or unsaturated), functional groups (alcohols, ketones, aldehydes, etc.) or individual species. Of the latter, benzene, toluene, ethylbenzene and xylene are collectively expressed as BTEX, these chemicals being ubiquitous in PW. Polycyclic aromatic hydrocarbons (PAHs) are also expressed as a single group of compounds, their toxicity having been long recognized [33].

Another key component of the organic fraction is the additives (Figure 1), synthetic compounds added to the injection water to assist its flow through the formation and suppress blockages. The exact composition with reference to the additives is not known and/or considered proprietary by the industry.

### 2.2. Refinery Effluent (RE)

Refineries generate products from crude oil (or “crude”) by thermal fractionation: separation of the crude constituents takes place by virtue of their differing boiling points. Wastewater streams generated from the refining process include (Table 2):tank bottom drawsdesalter effluentstripped sour water, andcooling tower blowdown.

Entrained water in crude originates from the oil well extraction process and/or from ingress during transportation [34]. It is usually removed as storage tank bottom sediment and water (BS and W) or by the desalter—a key component of the crude oil processing at the refinery—and forms part of the wastewater. A significant effluent stream derives from where pre-softened or stripped sour water has been in contact with hydrocarbons. Wastewaters generated from operations from where no direct contact with hydrocarbons arises include residual water rejected from boiler feedwater pre-treatment processes, water produced from: (i) regeneration of ion exchange resins in zeolite softeners and demineralisers, and (ii) blowdown (the concentrate stream) from cooling towers and boilers. There is also likely to be minor contamination of stormwaters from run off, as well as minor flows from laboratory discharges, washing and sewage.

The principal water stream in a refinery is the cooling water (CW), which makes up 50–55% of all the water in a refinery [34]. At times CW can bypass the WwTP to reduce its hydraulic loading provided the CW quality is appropriate for discharge. In addition, CW may be used for dilution of high-COD waters if they are otherwise bypassing the WwTP.

Since oil refining combines a number of different processes (Table 2) that generate effluents of different qualities, reported refinery effluent composition from different studies vary widely (Table 3). Temporal variations in effluent quality are significant, according to the sequencing of the discharges from the various internal operations. Consequently, the key reported determinant of COD varies by more than an order of magnitude—from around 200 to more than 5000 mg/L—across the different studies.

Notwithstanding the temporal and site/installation-related fluctuations in refinery effluent quality, a review of the data sets given in Table 1 and Table 3 indicates the PW and RE to be broadly similar in terms of the COD and oil content. However, it cannot necessarily be inferred from these data that the two streams are of comparable treatability.

## 3. Abiotic Low-Pressure (UF/MF) Membrane Separation

UF membranes have been used for treating small volumes of oil-laden industrial wastewaters since the mid 1970s [44], primarily to reduce the volume of wastewater to be disposed of off-site. Research into filtration of oil-laden waters by low-pressure membranes has subsequently been based on:bench-scale studies,crossflow operation of tubular or square-channel membrane elements to sustain high shears and thus suppress membrane surface fouling,analogue (or synthetic) effluent feeds,refinery wastewaters, andlimited duration (<6 h) trials conducted under constant transmembrane pressure (TMP) conditions.

Studies have demonstrated the expected effective rejection of emulsified and suspended oil down to levels well below legislated discharge limits [9,10,11,12,13,14,15,16,17,18,19,20,21,22,23], which are generally in the region of 30 mg/L total oil. This is currently the only stipulated water quality requirement for PW discharged from oil platforms, although there is a legal requirement for assessing environmental risk. Unlike the classical PW clarification technologies (Figure 1), membrane separation is not limited in efficacy by the oil droplet size.

The literature reveals an increased interest in ceramic membranes for this duty, with some commercial suppliers apparently collaborating in site-based demonstration trials [14,15]—though such trials have been very limited in number in the case of PW treatment. Ceramic materials provide greater tolerance to aggressive chemical and thermal conditions. They are also considered to offer greater resistance to fouling by the effluent hydrophobic content. Trials encompassing both real PW and its analogues have demonstrated the significantly greater fouling propensity of the former [10,11,19].

Outputs of the various of studies (Table 4) have been largely defined by a rapid decline in flux (or permeability) to some neo-steady-state value. There is no apparent pattern in this decline across the different studies, though it must clearly relate to key factors such as feedwater composition, hydraulics (primarily crossflow velocity, CFV) and temperature, as well as the characteristics of the membrane itself. Various studies [11,12,13,19,21,22] have demonstrated the efficacy of optimizing the physical (backflushing) and chemical cleaning to sustain the flux.

A key reported observation regarding the application of low-pressure membrane technology to PW concerns the impact of polymers, employed in chemical enhanced oil recovery (CEOR) [23,45]. Polymers are used to enhance the displacement of oil from the formation, but in doing so increase the PW viscosity and decrease the oil droplet size. The increased degree of emulsification of the oil challenges oil–water separation by the conventional hydrocyclone and induced gas flotation methods (Figure 2).

Critically, the polymer has been demonstrated to cause rapid fouling of ceramic membranes [23]. Given that the implementation of membrane technology for PW treatment on offshore platforms has been reported to rely on sustaining a flux above ~650 LMH [18,20] to ensure a sufficiently low footprint, the challenge imposed by polymer fouling associated with CEOR is significant. Whilst the fouling can be expected to be controlled by the backflush and chemical cleaning cycles, these increase the process downtime (and so decrease the net flux) and add to the installation footprint through the tankage and equipment requirement associated with storage and conveying of cleaning chemicals.

## 4. Membrane Biological Treatment (MBRs)

In contrast to the apparent absence of implemented UF/MF polishing of PW on offshore oil platforms, MBRs have been employed for treating refinery and petrochemical effluents for more than 15 years. An early example is the landmark Syndial plant at Porto Marghera in Italy, which treats an average daily flow of 38,400 m^3^/d and was installed in 2005. The largest congregation of petrochemical industry MBR plants is in China: by the end of 2010 there were at least 12 MBRs treating petroleum effluents, each of more than 5000 m^3^/d individual capacity, providing a combined treatment capacity of more than 130,000 m^3^/d in mainland China [46,47]. Against this, the implementation of MBRs for on-shore PW treatment appears to have been much more limited, with studies largely restricted to short-term bench-scale studies [29].

A crucial facet of the MBR technology, and biological processes generally, is that they provide removal of both dissolved and suspended organic material rather than just the suspended oil. A review of data reported for COD removal from refinery and petrochemical effluents based on both bench and pilot scale studies [48,49,50,51,52] and full-scale references [24,25,26] (Table 5) reveals them to achieve an average of 91% COD removal, leaving a residual of 56 mg/L on average (Figure 4).

Unlike the abiotic UF-MF application to PW treatment, where a very significant range of pseudo-steady state fluxes have been reported from bench and pilot scale studies (Table 4), the range of sustainable fluxes reported from full-scale MBR installations treating refinery and petrochemical effluents is relatively narrow at 9–17 LMH in the case of the immersed process configuration (iMBR), where the membrane modules are submerged in a tank (Figure 3b).

In the case of the sidestream configuration (sMBR), where the sludge from the bioreactor is pumped under pressure through an external multitube (MT) module (Figure 3c), the flux is significantly higher. The example given in Table 5 is the installation at Yanan Fengfuchuan, where a mean net flux of 44 LMH is reported based on the stated flow capacity and total membrane area. The sMBR configuration more closely resembles that of the PW membrane filtration plants (Figure 3a) and yields a commensurately higher flux than the immersed configuration due to the higher shears and TMPs applied.

As with abiotic membrane filtration of PW, MBR treatment of RE is often immediately preceded by flotation to reduce the load of free oil onto the membrane separation process, though sedimentation pretreatment appears to be favored for petrochemical effluents in China [47]. In the case of MBRs, dissolved air flotation (DAF) is employed rather than IGF: IGF is used for offshore PW treatment since oxygen has to be excluded from the treatment train to avoid corrosion issues and suppress explosion risks [6].

Moreover, MBRs are more tolerant of feedwater free oil than the abiotic process since the effluent flows initially into the biological process tank (Figure 3b,c). This tank contains a high concentration (8–10 g/L) of mixed liquor suspended solids (MLSS), which also retain the bacteria responsible for biodegrading the organic matter. Fouling of the membrane directly by the free oil is thus mitigated by: (a) the partitioning of the oil between the MLSS and the liquid phase, and (b) biodegradation by the bacteria. The low design flux of the iMBRs also significantly reduces membrane fouling propensity.

Whilst most MBR membrane products are polymeric, and more than half of these polyvinylidene difluoride (PVDF) [24], implementation of ceramic multitube membranes for potable water treatment began in the late 90s [53] and ceramic MBR membrane implementation has been steadily increasing since the first installation in the mid-noughties [54]. There are currently at least five commercially-available ceramic iFS module products [55]. Whilst the material is 4–6 times more costly than the polymeric membranes in terms of the purchase price per m^2^ membrane area, the membrane life is envisaged as being at least double that of the polymeric materials and the fluxes sustained significantly higher—partly because the mechanical strength and chemical resistance of the ceramic material permits more aggressive chemical cleaning [56].

## 5. Conclusions

Some key observations can be made regarding the relative extent of implementation of membrane technology for offshore PW treatment and onshore refinery/petrochemical effluent treatment:PW UF/MF membrane filtration studies have been limited in scale and duration, and largely based on synthetic/analogue feedwaters, which are not necessarily representative of real effluents. A wide range of final fluxes (4–700 LMH) and permeabilities (5–1240 LMH/bar) have consequently been reported, which may not be representative of full-scale operation.The most economical method of enhanced oil recovery (EOR) is through dosing with polymer. Whilst this improves the yield of oil from reservoirs, it also causes significant membrane fouling [23,45]. This decreases the flux, commensurately increasing the required membrane area and associated footprint to beyond the threshold where implementation on offshore oil platforms can be considered feasible [20,22].Ceramic membranes have been successfully implemented for onshore applications, with examples from 1997 onwards of abiotic potable water installations in Japan, [53] and, from 2007 onwards, MBR technologies for wastewater treatment [54]. Despite the apparent viability of ceramic membranes for these onshore duties, there has been no significant implementation for the key offshore application of PW filtration using either ceramic or polymeric membrane materials and no successful demonstration-scale trials reported.Onshore treatment of refinery and petrochemical effluents specifically using MBR technology, providing advanced biological treatment, has been established since the early noughties. Such treatment is appropriate since removal of both the suspended and dissolved organic matter (manifested as the COD) for this duty rather than just suspended oil removal as for PW treatment.The main contributors to MBR operational costs have been shown to be labor effort, sustainable flux, energy, and membrane replacement [48]. The comparative economic viability of ceramic membrane-based MBR technology for effluent treatment compared with the polymeric materials is thus dependent on the cost benefit offered by the reduced labor effort, longer membrane life and higher fluxes weighed against the cost penalty of the membrane material, as demonstrated by a recent review of potable water applications [55].

A simple SWOT analysis of the two technologies and applications (Figure 5) suggests that CEOR represents a key threat to the implementation of UF/MF offshore, unless the noted fouling challenge can be mitigated. Furthermore, the absence of a successful extended site-based demonstration of the technology remains a significant barrier. Against this, membrane filtration is perhaps the only reasonable option for reinjection of the recovered PW in the reservoir, given the severe economic ramifications of impairing the reservoir permeability if the water is insufficiently clarified.

MBR technology has been applied to oil industry effluents for over 15 years. It is particularly favored if water reuse is a key objective, since the process provides robustness pretreatment upstream of reverse osmosis demineralization. It is nonetheless relatively high in energy consumption, and the membrane replacement (in the case of polymeric membrane materials) adds significantly to the operating costs. The drive towards low-energy treatment solutions could present a significant threat to MBR technology implementation in the future, though energy efficiencies and general robustness of the process continue to improve.

It is concluded that the prospects of MBRs for onshore petroleum industry effluent treatment appear favorable. The implementation of abiotic UF/MF for offshore duties, on the other hand, appears to be hampered by the challenge imposed by membrane fouling and the associated impaired membrane permeability.

## Figures and Tables

**Figure 1 membranes-12-00391-f001:**
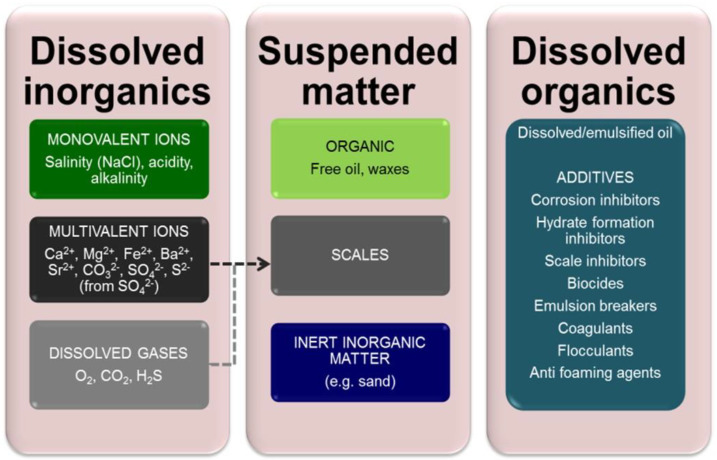
Produced water (PW) primary constituents.

**Figure 2 membranes-12-00391-f002:**
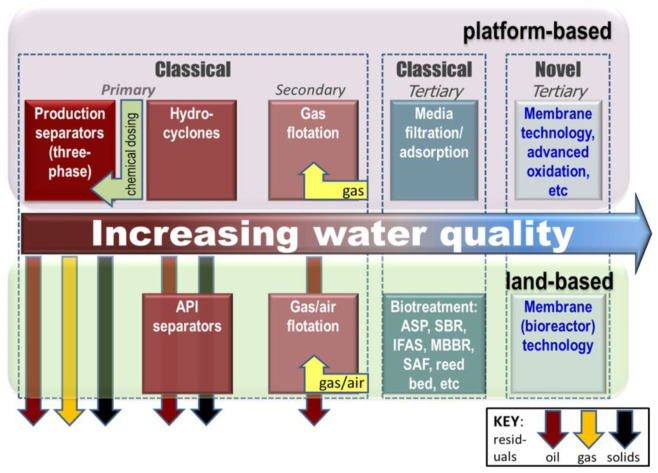
Treatment options offshore/platform-based (for PW) and onshore (PW and refinery/petrochemical effluents).

**Figure 3 membranes-12-00391-f003:**
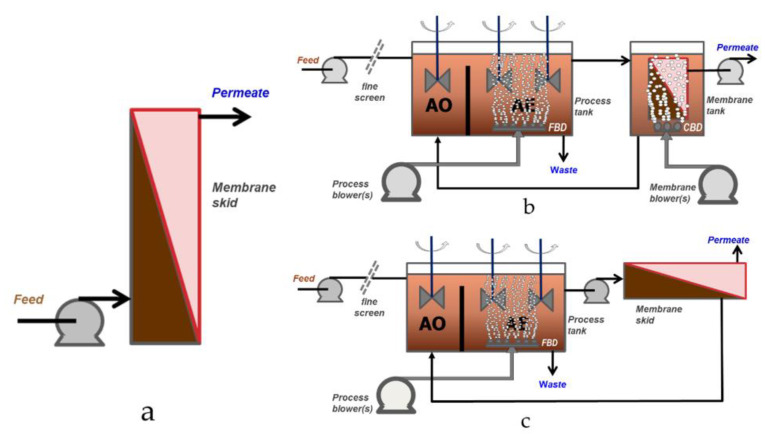
Membrane process schematics: (**a**) abiotic membrane filtration for PW treatment, (**b**) immersed membrane bioreactor (iMBR), and (**c**) sidestream membrane bioreactor (sMBR). FBD and CBD denote fine bubble and course bubble aerators. AO and AE denote the anoxic and aerobic zones of the bioreactor.

**Figure 4 membranes-12-00391-f004:**
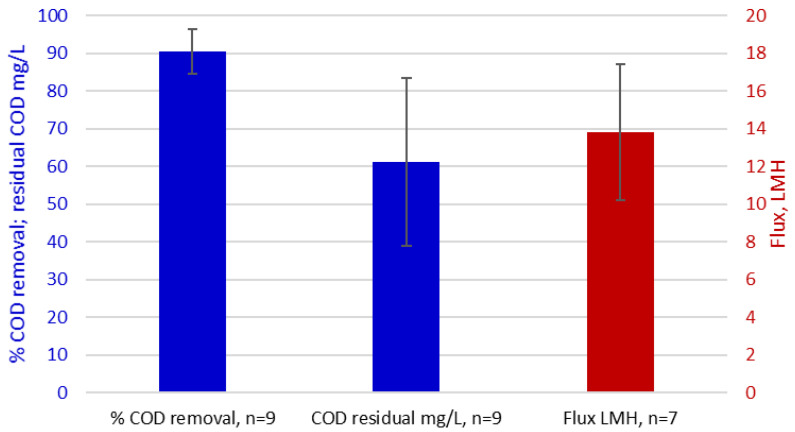
Mean key MBR process performance parameters, each based on 7–9 data points, for refinery and petrochemical treatment [48,49,50,51,52]. Bars represent the mean values of parameters abstracted from Table, and the error bars the standard deviation around the mean.

**Figure 5 membranes-12-00391-f005:**
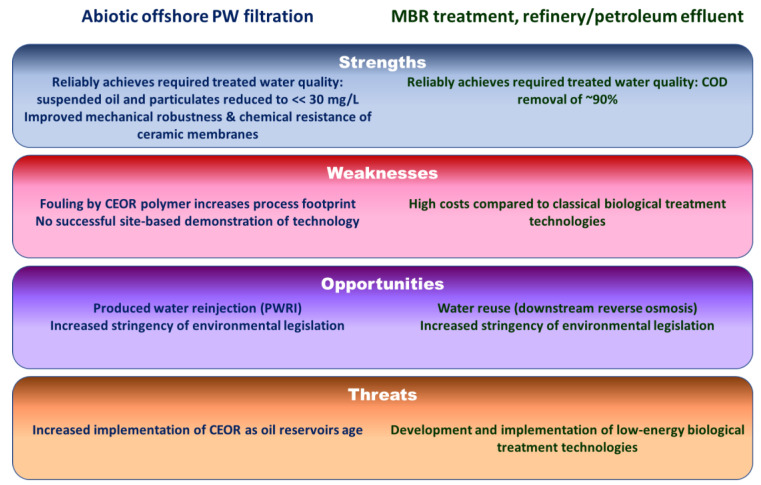
SWOT analysis, abiotic offshore PW membrane filtration vs. onshore MBR treatment of refinery and petroleum effluents.

**Table 1 membranes-12-00391-t001:** Oil field PW quality [31,32].

Parameter	Units	Min	Max
Density	kg/m^3^	1014	1140
Conductivity	μS/cm	4200	58,600
Salinity	mg/L	1000	>300,000
Total organic carbon, TOC	mg/L	-	1500
Chemical oxygen demand, COD	mg/L	1220	2600
BTEX ^a^	mg/L	0.7	24
Oil and grease	mg/L	2	565
Total suspended solids, TSS	mg/L	1.2	1000
Total dissolved solids, TDS	mg/L	100	400,000

^a^ benzene, toluene, ethylbenzene and xylene.

**Table 2 membranes-12-00391-t002:** RE stream water quality, mg/L [24].

Parameter	BS and W ^a^	Desalter	Stripped Sour Water	Cooling Tower Blowdown
COD	400–1000	400–1000	600–1200	150
Free hydrocarbons	Up to 1000	Up to 1000	<10	<5
SS	Up to 500	Up to 500	<10	Up to 200
Phenol	-	10–100	Up to 200	-
Benzene	-	5–15	negligible	-
Sulphides	Up to 100	Up to 100	-	-
Ammonia	-	Up to 100	-	-
TDS	High	High	Low	Intermediate

^a^ bottom sediment and water.

**Table 3 membranes-12-00391-t003:** Reported discharged RE water quality, mg/L.

Parameter	[35]	[36]	[37]	[38]	[39]	[40]	[41]	[42]	[43]
pH	8.3–8.9	6.3	8	7.5–10.3	6.7–8.2	7–9	5.6–6.0	8.0–8.2	-
BOD	-	61.4	950	-	300–630	150–350	65–80	570	150–350
COD	3600–5300	209	4800	330–556	2500–4100	300–600	228–481	850–1020	300–800
Phenol	11–14	-	-	-	-	-	-	98–128	20–200
Oil	160–185	11.3	-	40–91	50–100	50	76–105	12.7	3000
TPH	1.8–1.85	-	320	-	-	-	-	-	-
TOC	-	-	-	57–126	1290–2360	-	76–105	-	-
TSS	0.03–0.04	33.1	-	130–250	-	150	90–180	nd	100
BTEX	-	-	-	57–126	-	-	-	23.9	1–100
Sulphides	-	-	-	-	-	-	-	15–23	-
NH_3_	-	11.9	-	33–41	-	10–30	82	2.1–5.1	-

**Table 4 membranes-12-00391-t004:** Reported filtration performance from abiotic UF-MF studies.

Oil Concn mg/L, Water Source	Scale	Material	Pore Size μm	Init Flux, LMH	Fin Flux, LMH	TMP, bar	Fin Perm, LMH/bar	Ref
6000, RE	b	ZrO	0.2	240	120–175	1.1	109–159	[9]
366, PW	b	PS	0.007	225	128	1–1.7	95	[10]
	b	PS	0.006	100	70	1–1.7	74	
200–1000, tank dewatering effl.	b	AlO	0.2	128	28	1	28	[11]
	b	TiO	0.05	80	4	1	<5	
	b	TiO	0.05	120	30	1	120–30	
50–350, synth	p	AlO	0.1–0.5	-	80–175	0.06–0.25	400–800	[12]
Gas field PW	p	ZrO	0.05	-	170–255	-	190–250	[13]
3–25 (24–74), PW	p	SiC	0.1–0.5	25–120	50	0.3–1.5	150	[14]
PW	p	TiO	0.01–0.1	200	-	0.5–3.5	60	[15]
		SiC				0.25–1.5	135	
221–722, PW	p	SiC	0.04–0.1	-	135–590	0.35–0.95	450–1020	[16]
20, PW	p					0.6	520	
3000, RE	b	PS	0.1–0.2	145	65	1.5	50–15	[17]
SAGD effl.	p	AlO	0.05	200	-	1.52	132	[18]
	p	ZrO	202		-	1.52	45	
100, synth	b, p	ZrO	0.1	910	194–240	2	97–120	[19]
~60, RE	b, p	ZrO	0.1	910	175	2	88	
~250, RE	b	ZrO	0.1	1000	290	1.5	193	[20]
9–43, PW	p	AlO	0.2	-	295–312	2.5	118–125	[21]
	b	PAN	202	-	104–280	5	20–36	
24–95, PW	p	SiC	0.04	-	~350	0.55–0.6	370–380	[22]
			0.5	-	~700		1190–1240	
100, synth + poly	b	ZrO	0.1	430	69	1–2	35	[23]

LMH liters per m^2^ per hr; SAGD Steam-assisted gravity drainage; OFPW oilfield produced water; TMP transmembrane pressure; TSS total suspended solids; b bench; p pilot; synth synthetic (analogue) feed; AlO aluminium oxide; PS Polysulphone; SiC silicon carbide; TiO titanium oxide; ZrO zirconium oxide; poly CEOR polymer.

**Table 5 membranes-12-00391-t005:** Reported performance of full-scale MBR installations treating refinery and petrochemical effluents.

	Porta Maghera	Sinopec Guangzhou	Sinopec Jinling	Yunlin/Formosa	Dalsung	Petrobras [24]	Yanan Fengfuchuan
Country	Italy	China	China	Taiwan	Korea	Brazil	China
Application	Petrochemical	Refinery	Petroleum	Petroleum	Industrial park	Refinery	Oilfield reinjection
Configuration	iHF	iHF	iHF	iHF	iHF	iHF	sMT
Material	PVDF	PVDF	PVDF	PVDF	HDPE	PVDF	PES
Selection reason(s)	Required discharge WQ	Re-use (cooling towers)	Re-use (cooling towers)	Restricted footprint and water re-use.	Required discharge WQ and restricted footprint.	Required discharge WQ	Limit risk of reservoir pore plugging
Capacity (MLD)	47.5 PDF	4.8 ADF	6 ADF	25 PDF	25 PDF	7.2 PDF	1.5 PDF
HRT, h	20	18	16	-	8	-	-
SRT, d	-	49	26	-	45	36	-
Feed COD, mg/L	280	~150	~300	990	50	850	45 (oil)
% COD rem	>96%	>80%	>80%	95%	>80%	93%	>98% (oil)
MLSS, g/L	8.4 (design)	3 (3.5 MT)	3 (4.5, MT)	3.5	8	10	-
Flux (LMH)	9	9	10	17	12.5	16.4	44
SEC_m_ (kWh/m^3^)	-	0.60	0.60	-	0.70	-	-
Notes	Pre and post DN employed	MBR downstream of DAF + oxidation ditch. 12 h EQ	MBR downstream of primary sed. 40 h EQ	Probably downstream of primary sedimentation	30% sewage feed. MBR downstream of primary sedimentation	75% sewage feed. 25% industrial stream pre-clarified and w. 8 h EQ	MBR downstream of skimmer and DAF.

iHF immersed hollow fiber; sMT sidestream multitube; PVDF polyvinylidene difluoride; HDPE high-density polyethylene; PES polyethylsulphone; WQ water quality; MLD megaliters per day; ADF average daily flow; PDF peak daily flow; HRT hydraulic retention time; SRT solids retention time; MLSS mixed liquor suspended solids; SED_m_ specific energy consumption of membrane permeation; DN denitrification; DAF dissolved air flotation; EQ equalization.

## Data Availability

Not applicable.

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
