# Peer review of "A Brief Review of the Status of Low-Pressure Membrane Technology Implementation for Petroleum Industry Effluent Treatment"

_membranes, 2022, doi:10.3390/membranes12040391_

Round 1

Reviewer 1 Report

In this review, the authors realized a brief analysis of the current status of implementation of low-pressure membrane technology for the treatment of effluents in the petroleum industry. The text is well organized and contains a lot of data acquired from studies involving real effluents and large-scale installations. There are just a few minor aspects that should be taken into consideration before publication.

  1. The introduction should contain more theoretical information about onshore and offshore effluent treatment facilities (e.g. definition, examples, characteristics, etc) with appropriate references.
  2. Since there are a lot of abbreviations used in the manuscript, a list/table of abbreviations should be included at the beginning.
  3. Lines 27-30, the sentence is unclear, it should be reformulated with more accessible terms.
  4. The text should be re-read to check for typos (e.g. line 91 “formation”) and ensure all sentences are clear and concise.
  5. I noticed that there are a few paragraphs that don’t include references (e.g. lines 87-101, 118-132, 159-164, etc). The sources where the information was acquired must always be mentioned in the text.
  6. Considering its structure and contents, chapter 5 could be named “Conclusions”.

Author Response

In this review, the authors realized a brief analysis of the current status of implementation of low-pressure membrane technology for the treatment of effluents in the petroleum industry. The text is well organized and contains a lot of data acquired from studies involving real effluents and large-scale installations. There are just a few minor aspects that should be taken into consideration before publication.

  1. The introduction should contain more theoretical information about onshore and offshore effluent treatment facilities (e.g. definition, examples, characteristics, etc) with appropriate references.

Produced water is now defined within the first paragraph of Section 1 in the revised manuscript as follows: Produced water (PW) is water which is extracted from an oil well with the crude oil during crude oil production. Examples and characteristics are provided in Section 2.1, with the range of constituent concentrations given in Table 1.

Refinery and petrochemical effluents are defined at the start of the second paragraph of Section 1 in the revised manuscript as follows: Key onshore petroleum effluents comprise those discharged from refining and petrochemical manufacturing. Refining refers to the fractionation of crude oil into primarily transportation fuels, along heating oils and other more minor oil elements. Petrochemicals are the intermediates used to produce industrial and consumer products (e.g. plastics, rubbers, resins, synthetic fibers, adhesives, dyes, detergents and pesticides).

  1. Since there are a lot of abbreviations used in the manuscript, a list/table of abbreviations should be included at the beginning.

This is now included in the revised manuscript.

  1. Lines 27-30, the sentence is unclear, it should be reformulated with more accessible terms.

PW is now defined in the previous sentence (see #1 above). The sentence has been rewritten and simplified: .. results from displacing oil in the reservoir with environmental waters – most often seawater – and contains many chemical species which are onerous to the environment and/or challenging to oil abstraction. They include cations which form scales, such as calcium and magnesium, and toxic species such as hydrogen sulphide and heavy metals.

  1. The text should be re-read to check for typos (e.g. line 91 “formation”) and ensure all sentences are clear and concise.

The entire revised manuscript has been carefully proofread by a native English speaker. The above typo has been corrected in the revised manuscript.

  1. I noticed that there are a few paragraphs that don’t include references (e.g. lines 87-101, 118-132, 159-164, etc). The sources where the information was acquired must always be mentioned in the text.

References have been added at the above junctures, as well as a few others in the revised manuscript.

  1. Considering its structure and contents, chapter 5 could be named “Conclusions”.

Section 5 has been renamed as suggested in the revised manuscript.

Reviewer 2 Report

I have reviewed this manuscript entitled "A brief review of the status of low-pressure membrane technology implementation for petroleum industry effluent treatment". This manuscript aims to summarize the status of abiotic UF-MF for offshore PW treatment and onshore refinery/petrochemical effluent treatment. The topic is meaningful. The manuscript can be improved.

My detailed comments:

  1. Abstract need be enhanced it has to address briefly and describe aim, object, procedure, results and important findings of novelty accordingly. The current abstract is not informative enough and should be rewritten more scientifically.

  1. Please provide a nomenclature list, a glossary of any symbols of abbreviations used in the text. Also, all of the acronym should be expanded first before using it as an acronym.
  2. Page 1 Line nos. 27-30, recent studies on PW treatment in EOR process of oilfield can be read and enrich your statement, "Foaming Properties and Foam Structure of Produced Liquid in Alkali/Surfactant/Polymer Flooding Production. Journal of Energy Resources Technology, 2021, 143(10): 103005", and “Polymer-Flood Produced-Water-Treatment Trials(2014). Oil and Gas Facilities, 3 (6): 89-100”.

  1. Section 2: the authors should discuss the requirements of such water quality on treatment process. Moreover, is this water quality representative?

  1. -It is not clear on which basis, figure 4 was formed. It needs to be explained.

  1. Please provide Conclusions of your review.

  1. Results must be discussed relevant to the opportunity and application. How about the technoeconomic status of the MBRs and potential effect on conventional treatment process.

  2. Please improve the Figures. I believe the quality of figures does not meet the publication requirement of the host journal.

Author Response

I have reviewed this manuscript entitled "A brief review of the status of low-pressure membrane technology implementation for petroleum industry effluent treatment". This manuscript aims to summarize the status of abiotic UF-MF for offshore PW treatment and onshore refinery/petrochemical effluent treatment. The topic is meaningful. The manuscript can be improved.

  1. Abstract need be enhanced it has to address briefly and describe aim, object, procedure, results and important findings of novelty accordingly. The current abstract is not informative enough and should be rewritten more scientifically.

The abstract has now been rewritten and expanded to include the subject background, a definition of the aim, the scope of the information and data provided, and key outputs and novel findings of the article.

  1. Please provide a nomenclature list, a glossary of any symbols of abbreviations used in the text. Also, all of the acronym should be expanded first before using it as an acronym.

A definition of all abbreviations is included in the revised article (see also Reviewer 1, point #2). All abbreviations (including acronyms) are now defined when first introduced in the text, or else in the table footnotes.

  1. Page 1 Line nos. 27-30, recent studies on PW treatment in EOR process of oilfield can be read and enrich your statement, "Foaming Properties and Foam Structure of Produced Liquid in Alkali/Surfactant/Polymer Flooding Production. Journal of Energy Resources Technology, 2021, 143(10): 103005", and “Polymer-Flood Produced-Water-Treatment Trials(2014). Oil and Gas Facilities, 3 (6): 89-100”.

These references (58-59) have now been cited and added to the reference list.

  1. Section 2: the authors should discuss the requirements of such water quality on treatment process. Moreover, is this water quality representative?

The revised text now includes the following sentence at the start of Section 2 regarding the water quality parameters targeted by the treatment scheme: Different treated water quality objectives apply to PW for sea discharge and land-based petroleum effluents. For the former, the discharged water quality is based on the oil concentration, and physicochemical removal of the suspended oil is normally sufficient to meet the requirement. The required water quality for effluents discharged inland, on the other hand, is normally based on the total organic concentration measured as the COD or BOD.

With regards to representativeness, the variability of water quality is already acknowledged. The text in Section 2.1 includes the sentence: The ranges of concentration of the key PW bulk water quality parameters (Table 1) vary widely according to the reservoir formation and other geological characteristics [2-6]. A similar caveat has been added to the refinery effluent section (Section 2.2): Since the oil refining combines a number of different processes (Table 2) which generate effluents of different qualities, reported refinery effluent composition from different studies vary widely (Table 3). The sources of the water quality data are cited in the article.

  1. It is not clear on which basis, figure 4 was formed. It needs to be explained.

The following text has been added to the caption: Bars represent the mean values of parameters abstracted from Table 5, and the error bars the standard deviation around the mean.

  1. Please provide Conclusions of your review.

The final section has been renamed “Conclusions”, at the suggestion of Reviewer 1 (see point #6).

  1. Results must be discussed relevant to the opportunity and application. How about the technoeconomic status of the MBRs and potential effect on conventional treatment process.

Comparative technoeconomic assessments of MBRs and conventional activated sludge (CAS) have been published elsewhere, most recent examples including:

  • Gao et al (2022). Techno-economic characteristics of wastewater treatment plants retrofitted from the conventional activated sludge process to the membrane bioreactor process. Env. Sci. Engng., 16(4);
  • Gao et al (2021). Cost-benefit analysis and technical efficiency evaluation of full-scale membrane bioreactors for wastewater treatment using economic approaches. Clean. Prodn., 301 126984;
  • Dalri-Cecato et al (2020). CAPEX and OPEX evaluation of a membrane bioreactor aiming at water reuse. Sci. Engng., 47-63.

These have all tended to indicate a cost benefit for the MBR technology, mirroring the conclusions from the earlier landmark study of Young et al (2012), MBR vs. CAS: capital and operating cost evaluation, Water Pract. Technol. 7(4). However, these studies have all been based on municipal effluent treatment. According to Zhang et al (ref 44 in the current paper), the average investment value for industrial effluent treatment by MBRs is almost double that for treating municipal effluent, but there is no corresponding comparison for conventional biological treatment of industrial wastewaters. Thus, the technoeconomic status of MBRs in comparison to the conventional treatment process appears not to have been examined for industrial effluents. Conducting such an analysis would certainly be a worthwhile exercise but would also constitute a paper in its own right.

  1. Please improve the Figures. I believe the quality of figures does not meet the publication requirement of the host journal.

This point has been referred to the journal editorial: any required improvement in the graphical quality will be attended to. 

Round 2

Reviewer 2 Report

1.The conclusion was well summarized, but it can be briefed up.

2.The added literatures did not cited in the text.

Author Response

See annotated text in green in revised manuscript (file attached)

The mis-citation of the two new references has now been corrected (Lines 58-59)

The Conclusions have been reduced in word count from 608 to ~550 words.